# Learning to Tokenize for Generative Retrieval

**Weiwei Sun**[1], **Lingyong Yan**[2], **Zheng Chen**[1], **Shuaiqiang Wang**[2], **Haichao Zhu**[2]
**Pengjie Ren**[1], **Zhumin Chen**[1], **Dawei Yin**[2], **Maarten de Rijke**[3], **Zhaochun Ren**[4*]
[1]Shandong University, China    [2]Baidu Inc., China
[3]University of Amsterdam, The Netherlands    [4]Leiden University, The Netherlands
{sunnweiwei,lingyongy}@gmail.com   yindawei@acm.org
m.derijke@uva.nl   z.ren@liacs.leidenuniv.nl

## Abstract

As a new paradigm in information retrieval, generative retrieval directly generates a ranked list of document identifiers (docids) for a given query using generative language models (LMs). How to assign each document a unique docid (denoted as *document tokenization*) is a critical problem, because it determines whether the generative retrieval model can precisely retrieve any document by simply decoding its docid. Most existing methods adopt rule-based tokenization, which is ad-hoc and does not generalize well. In contrast, in this paper we propose a novel document tokenization learning method, GENRET, which learns to encode the complete document semantics into docids. GENRET learns to tokenize documents into short discrete representations (i.e., docids) via a discrete auto-encoding approach. We develop a progressive training scheme to capture the autoregressive nature of docids and diverse clustering techniques to stabilize the training process. Based on the semantic-embedded docids of any set of documents, the generative retrieval model can learn to generate the most relevant docid only according to the docids' semantic relevance to the queries. We conduct experiments on the NQ320K, MS MARCO, and BEIR datasets. GENRET establishes the new state-of-the-art on the NQ320K dataset. Compared to generative retrieval baselines, GENRET can achieve significant improvements on unseen documents. Moreover, GENRET can also outperform comparable baselines on MS MARCO and BEIR, demonstrating the method's generalizability.

## 1 Introduction

Document retrieval plays an essential role in web search applications and various downstream knowledge-intensive tasks by identifying relevant documents to satisfy users' queries. Recently, *generative retrieval* has emerged as a new paradigm for document retrieval [1, 5, 37, 41, 46, 47] that directly generates a ranked list of document identifiers (docids) for a given query using generative language models (LMs). Unlike dense retrieval [9, 13, 23, 42], generative retrieval presents an end-to-end solution for document retrieval tasks [37]. It also offers a promising approach to better exploit the capabilities of recent large LMs [1, 41].

As shown in Figure 1, document tokenization aims to tokenize each document in the corpus as a sequence of discrete characters, i.e., docids. Document tokenization plays a crucial role in generative retrieval, as it defines how the document is distributed in the semantic space [37]. And it is still an open problem how to define docids. Most previous generative methods tend to employ rule-based document tokenizers, such as generating titles or URLs [5, 46], or clustering results from off-the-shelf document embeddings [37, 41]. Such rule-based methods are usually ad-hoc and do not generalize well. In particular, the tokenization results potentially perform well on retrieving documents that have been seen during training, but generalize poorly to unlabeled documents [17, 20].

---

*Corresponding author.

37th Conference on Neural Information Processing Systems (NeurIPS 2023).

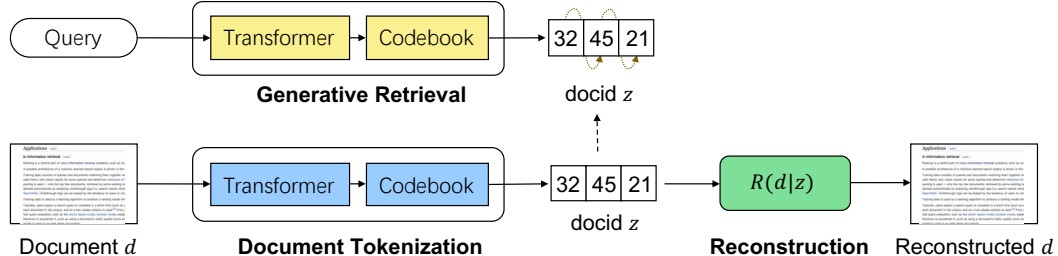

Figure 1: An overview of our proposed method. The proposed method utilizes a document tokenization model to convert a given document into a sequence of discrete tokens, referred to as a docid. This tokenization process allows for the reconstruction of the original document through a reconstruction model. Subsequently, an autoregressive generation model is employed to retrieve documents through the generation of their respective docids.

To address the above problem, we propose GENRET, a document tokenization learning framework that learns to tokenize a document into semantic docids in a discrete auto-encoding scheme. GEN-RET consists of a shared sequence-to-sequence-based document tokenization model, a generative retrieval model, and a reconstruction model. In the proposed auto-encoding learning scheme, the tokenization model learns to convert documents to discrete docids, which are subsequently utilized by the reconstruction model to reconstruct the original document. The generative retrieval model is trained to generate docids in an autoregressive manner for a given query. The above three models are optimized in an end-to-end fashion to achieve seamless integration.

There are usually two challenges when using auto-encoding to optimize a generative retrieval model: (i) docids with an autoregressive nature, and (ii) docids with diversity. To address the first challenge and also to stabilize the training of GENRET, we devise a progressive training scheme. This training scheme allows for a stable training of the model by fixing optimized prefix docids $z_{<t}$. To optimize the docids at each step, three proposed losses are utilized: (i) a reconstruction loss for predicting the document using the generated docid, (ii) a commitment loss for committing the docid and avoiding forgetting, and (iii) a retrieval loss for optimizing the retrieval performance end-to-end. To address the second challenge, we propose a parameter initialization strategy and a re-assignment of the docid based on a *diverse clustering* technique to increase the diversity of the generated docids.

We conduct extensive experiments on three well-known document retrieval benchmark datasets, NQ320K [15, 37], MS MARCO [4, 46], and BEIR [38]. GENRET attains superior retrieval performance against state-of-the-art generative retrieval models on NQ320K. GENRET achieves +14% relative improvements on the unseen test set of NQ320K over the best generative retrieval baseline. Experiments on MS MARCO and six BEIR datasets also show that GENRET outperforms existing generative methods and achieves competitive results compared to popular dense retrieval models. Experiments on retrieving new documents, analytical experiments, and an efficiency analysis confirm the effectiveness of the proposed model.

We summarize our contributions as follows: (i) We propose GENRET, a generative retrieval model that represents documents as discrete semantic docids. To the best of our knowledge, this is the first tokenization learning method for document retrieval. (ii) We propose an auto-encoding approach, where the docids generated by our tokenization model are reconstructed by a reconstruction model to ensure the docids capture the semantic information of the document. (iii) We devise a progressive training scheme to model the autoregressive nature of docids and stabilize the training process. (iv) Experimental results demonstrate that GENRET achieves significant improvements, especially on unseen documents, over generative retrieval baselines.[2]

## 2   Preliminaries

The document retrieval task can be formalized as the process of retrieving a relevant document $d$ for a search query $q$ from a collection of documents $\mathcal{D}$. Each document $d \in \mathcal{D}$ is assumed to be a plain text consisting of a sequence of tokens, denoted as $d = \{d_1, \ldots, d_{|d|}\}$, where $|d|$ is the total number of tokens in the document. For generative retrieval models, it is usually challenging and computationally

---

[2]The code of this work is available at `www.github.com/sunnweiwei/GenRet`.

inefficient to directly generate original documents of typically long length. Therefore, most existing approaches rely on a technique named **document tokenization**, which represents a document $d = \{d_1, \ldots, d_{|d|}\}$ as a shorter sequence of discrete tokens (docid) $z = \{z_1, \ldots, z_t, \ldots, z_M\}$, where each token $z_t$ is as a $K$-way categorical variable, with $z_t \in [1, 2, \ldots, K]$, and $M$ is the length of the docid. See Figure 1 for an example of document tokenization with $M = 3$ and $K = 64$.

As an alternative sequence of the original document, the tokenized docid $z$ should satisfy the following two properties: (i) different documents have short but different docids; and (ii) docids capture the semantics of their associated documents as much as possible [37]. Because $z$ is a sequence of a fixed length and usually shorter than the original document $d$, the model's training and inference can be simplified and more efficient. This paper employs a tokenization model $Q \colon d \to z$ to map $d$ to docid $z$. More details about $Q$ are provided in Section 3.1. After tokenizing each document to docid $z$, a generative retrieval model $P \colon q \to z$ learns to retrieve relevant documents by generating a query $q$ to a docid $z$ autoregressively [37].

## 3 Method

Conventionally, document tokenization is done by a fixed pre-processing step, such as using the title of a document or the results of hierarchical clustering obtained from BERT [5, 37]. However, it has been observed that such ad-hoc document tokenization methods often fail to capture the complete semantics of a document. For example, the title of a web page often does not exist or has low relevance to the content of the web page, and the use of clustering-based docids arbitrarily defines the document in discrete space.

In this paper, we propose GENRET, a novel tokenization learning method based on discrete auto-encoding, to learn semantic docid in a fully end-to-end manner. Figure 1 gives an overview of the proposed method. The proposed GENRET comprises three main components: (i) a sequence-to-sequence based retrieval model $P(z \mid q)$, (ii) a document tokenization model $Q(z \mid d)$, and (iii) a reconstruction model $R(d \mid z)$. The document tokenization model tokenizes a document $d$ into unique discrete variables $z$, and the retrieval model is trained to generate the latent variables $z$ for a given query $q$. In addition, the reconstruction model is used to re-generate the original document from $z$ to ensure $z$ captures the semantics of the original document as much as possible.

### 3.1 Model architecture

Following DSI [37], we employ an encoder-decoder Transformer to implement the generative retrieval model. Specifically, given an input text $d^3$, the T5-based tokenization model encodes $d$ and a prefix of docid $z_{<t}$ and continuously produces latent representation $\mathbf{d}_t$ of $d$ at time step $t$:

$$\mathbf{d}_t = \text{Decoder}(\text{Encoder}(d), z_{<t}) \in \mathbb{R}^D, \tag{1}$$

where $D$ denotes the hidden size of the model. $\text{Encoder}(d)$ denotes the output of the Encoder.

Then, the tokenization model generates a token for each document based on $\mathbf{d}_t$. At each timestep $t$, we define an external embedding matrix named *codebook* $\mathbf{E}_t \in \mathbb{R}^{K \times D}$, where $K$ is the size of the discrete latent space. There are $K$ embedding vectors $\mathbf{e}_{t,j} \in \mathbb{R}^D, j \in [K]$, and each vector $\mathbf{e}_{t,j}$ can be regarded as the centroid of a segmentation.

Based on the *codebook* $\mathbf{E}_t$, the discrete latent variable $z_t$ at timestep $t$ is calculated by a dot-product look-up using the codebook $\mathbf{E}_t$:

$$Q(z_t = j \mid z_{<t}, d) = \text{Softmax}_j(\mathbf{d}_t \cdot \mathbf{E}_t^\top), \tag{2}$$

where $Q(z_t = j \mid z_{<t}, d)$ denotes the probability of tokenizing $d$ to a particular value $j \in [K]$ at timestep $t$, $\text{Softmax}_j$ is a softmax function to output the probability of axis $j$.

**Document reconstruction model.** The docid generated by the tokenization model $Q$ is required to capture the semantic information of the document. To this end, we propose an auto-encoding training scheme, where a reconstruction model $R \colon z \to d$ that predicts $d$ using $z$ is designed to force the tokenization model $Q \colon d \to z$ to reproduce a docid $z$ that can be reconstructed back-to-the original document.

---

[3]We use document $d$ here for the denotation, noting that the computation is the same when $q$ is input.

The input of the reconstruction model is docid $z$, and the output is its associated document $d$. We first embed $z$ into representation matrix $\mathbf{z} = \{\mathbf{z}_1, \ldots, \mathbf{z}_M\} \in \mathbb{R}^{M \times D}$ using the codebook of the tokenization model:

$$\mathbf{z} = \{\mathbf{e}_{1,z_1}, \mathbf{e}_{2,z_2}, \ldots, \mathbf{e}_{M,z_M}\} \in \mathbb{R}^{M \times D}, \tag{3}$$

where each $t \in [M]$, $\mathbf{z}_t = \mathbf{e}_{t,z_t} \in \mathbb{R}^D$ is the embedding vector of $z_t$ in the $t$-step codebook $\mathbf{E}_t$.

We then devise a retrieval-based reconstruction model that predicts the target document $d$ by retrieving it from document collection $\mathcal{D}$, based on the inputs $\mathbf{z}$. The relevance score between the input docid $z$ and the target document $d$ is defined as follows:

$$R(d \mid \mathbf{z}) = \prod_{t=1}^{M} \frac{\exp(\mathbf{z}_t \cdot \mathrm{sg}(\mathbf{d}_t^\top))}{\sum_{d^* \in S(z_{<t})} \exp(\mathbf{z}_t \cdot \mathrm{sg}(\mathbf{d}^*{}_t^\top))}, \tag{4}$$

where $S(z_{<t})$ is a sub-collection of $\mathcal{D}$ consisting of documents that have a docid prefix that is the same as $z_{<t}$. $d^* \in S(z_{<t})$ represents a document from the sub-collection $S(z_{<t})$. $\mathbf{d}_t$ and $\mathbf{d}^*{}_t$ are continuous representations of documents $d$ and $d^*$, respectively, as defined in Eq. 1. The operator $\mathrm{sg}(\cdot)$ is the stop gradient operator to prevent gradient back propagation. Intuitively, $R(d \mid \mathbf{z})$ is designed to retrieve a specific document $d$ from a set of documents $S(z_{<t})$ at each timestep $t$. The set $S(z_{<t})$ only includes those documents that are assigned the same docid prefix $z_{<t}$ as the target document $d$. By utilizing this loss function, at each step $t$, the model is facilitated to learn the residual semantics of the documents not captured by the previous docid $z_{<t}$.

## 3.2 Model optimization

For the document tokenization model $Q(z \mid d)$, generative retrieval model $P(z \mid q)$, and reconstruction model $R(d \mid z)$, jointly optimizing these three models using auto-encoding is challenging due to the following reasons: (i) **Learning docids in an autoregressive fashion**. On one hand, the prediction of the $z_t$ at time $t$ relies on previously predicted docids $z_{<t}$, which is often under-optimized at the beginning and rapidly changes during training, making it difficult to reach convergence. On the other hand, simultaneously optimizing $z$ makes it challenging to guarantee a unique docid assignment. Hence, to stabilize the training of GENRET, we devise a *progressive training scheme* (see Section 3.2.1). (ii) **Generating docids with diversity**. Optimizing the model using auto-encoding often leads to unbalanced docid assignment: a few major docids are assigned to a large number of documents and most other docids are rarely assigned. Such a sub-optimal distribution of docids affects the model distinguishability, which in turns triggers length increments of docids in order to distinguish conflicting documents. We introduce two *diverse clustering* techniques to ensure docid diversity (see Section 3.2.2).

### 3.2.1 Progressive training scheme

To optimize each of the three models listed above in an autoregressive manner, we propose a progressive auto-encoding learning scheme, as illustrated in Figure 2. The whole learning scheme contains $M$ learning steps with respect to the final docid in $M$-token. And the docid $z_T$ at step $T \in [M]$ is learned and optimized at the corresponding learning step. Besides, at each step $T \in [M]$, the docid $z_T$ and the model parameters associated with $z_T$ generation are updated, while previously produced docids $z_{<T}$ and other parameters are kept fixed. By progressively performing the above process, we can finally optimize and learn our models.

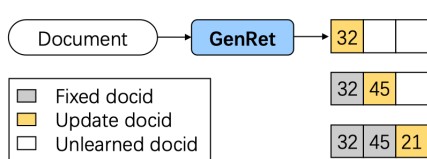

Figure 2: Progressive training scheme. $z_t$ (docid at timestep $t$) is optimized at the $t$-th training step, while $z_{<t}$ (docids before timestep $t$) are kept fixed.

At each optimization step, say the $T$-step, we devise the learning objective for document tokenization consisting of three loss functions detailed below.

**Reconstruction loss.** We utilize the reconstruction model $R(d \mid z)$ as an auxiliary model to learn to optimize the docid generation, whose main goal is capturing as much semantics in the docid as

possible. Therefore, we define a reconstruction loss function of step $T$ as follows:

$$\mathcal{L}_{\text{Rec}} = -\log R(d \mid \hat{\mathbf{z}}_{\leq T}), \quad \text{where } \hat{\mathbf{z}}_{\leq T} = \{\text{sg}(\mathbf{z}_1), \ldots, \text{sg}(\mathbf{z}_{T-1}), \mathbf{z}_T\} \in \mathbb{R}^{T \times D}$$

$$\forall t \in [T]: \mathbf{z}_t = \mathbf{e}_{t,j^*} \in \mathbb{R}^D, \quad \text{where } j^* = \arg\max_j Q(z_t = j \mid z_{<t}, d), \tag{5}$$

where $\hat{\mathbf{z}}_{\leq T}$ is the first $T$ representations of the $z$, and only the variable $\mathbf{z}_T$ is optimized in step $T$. $Q(z_t = j \mid z_{<t}, d)$ is defined in Eq. 2. And the document tokenization model $Q$ can therefore be optimized when minimizing $\mathcal{L}_{\text{Rec}}$.

Of note, since the computation involves a non-differentiable operation $(\arg\max(\cdot))$, we apply straight-through gradient estimation to back-propagate the gradient from reconstruction loss to $\mathbf{d}_T$ following [39, 44], which copies the gradient of $\mathbf{z}_T$ directly to $\mathbf{d}_T$. Specifically, the gradients to document representation $\mathbf{d}_T$ are defined as $\frac{\partial \mathcal{L}_{\text{Rec}}}{\partial \mathbf{d}_T} := \frac{\partial \mathcal{L}_{\text{Rec}}}{\partial \mathbf{z}_T}$. And the gradients to the *codebook* embedding $\mathbf{e}_{T,j}$ are defined as $\frac{\partial \mathcal{L}_{\text{Rec}}}{\partial \mathbf{e}_{T,j}} := 1_{z_T=j} \frac{\partial \mathcal{L}_{\text{Rec}}}{\partial \mathbf{z}_T}$.

**Commitment loss.** In addition, to make sure the predicted docid commits to an embedding and to avoid models forgetting previous docid $z_{<t}$, we add a commitment loss as follows:

$$\mathcal{L}_{\text{Com}} = -\sum_{t=1}^{T} \log Q(z_t \mid z_{<t}, d). \tag{6}$$

**Retrieval loss.** For the generative retrieval model $P$, we jointly learn it together with the document tokenization model $Q$, where $P$ learns to generate the docids of relevant documents $d$ given a query $q$. Specifically, suppose $(q, d)$ are a query and relevant document pair; we define the learning objective of retrieval model $P$ as:

$$\mathcal{L}_{\text{Ret}} = -\log \frac{\exp(\mathbf{q}_T \cdot \mathbf{d}_T)}{\sum_{d^* \sim B} \exp(\mathbf{q}_T \cdot \mathbf{d}^*_T)} - \sum_{t=1}^{T} \log P(z_t \mid z_{<t}, q), \tag{7}$$

where the first term is a ranking-oriented loss enhancing the model using $(q, d)$ pair; $d^*$ is an in-batch document sampled from the same training mini-batch $B$; $\mathbf{q}_T$ and $\mathbf{d}_T$ denote the representation of $q$ and $d$ at timestep $T$. The second term is the cross-entropy loss for generating docid $z$ based on $q$.

The final loss we use at step-$T$ is the sum of reconstruction loss, commitment loss, and retrieval loss:

$$\mathcal{L} = \mathcal{L}_{\text{Rec}} + \mathcal{L}_{\text{Com}} + \mathcal{L}_{\text{Ret}}. \tag{8}$$

### 3.2.2 Diverse clustering techniques

To ensure diversity of generated docids, we adopt two diverse clustering techniques–codebook initialization and docid re-assignment at each progressive training step, where codebook initialization mainly aims to increase the balance of semantic space segmentation, and the docid re-assignment mainly aims to increase the balance of docid assignments.

**Codebook initialization.** In order to initialize the codebook for our model, we first warm-up the model by passing the continuous representation $\mathbf{d}_T$ to the reconstruction model instead of the docid representation $\mathbf{z}_T$ as defined in Eq. 3. During this warm-up phase, we optimize the model using the reconstruction loss $\mathcal{L}_{\text{Rec}}$ and commitment loss $\mathcal{L}_{\text{Com}}$. Next, we collect the continuous representations $\mathbf{d}_T$ of all documents in $\mathcal{D}$, and cluster them into $K$ groups. The centroids of these clusters are then used as the initialized codebook $\mathbf{E}_T$. To balance the initialized docid distribution, we utilize a diverse constrained clustering algorithm, *Constrained K-Means*, which first normalizes the embeddings of each prefix group, and modifies the cluster assignment step (E in EM) by formulating it as a minimum cost flow (MCF) linear network optimization problem [2].

**Docid re-assignment.** In order to assign docids to a batch of documents, we modify the dot-product look-up results in Eq. 2 by ensuring that the docid for different documents in the batch are distinct following the method described in [6, 44]. Specifically, let $\mathbf{D}_t = \{\mathbf{d}_t^{(1)}, \ldots, \mathbf{d}_t^{(B)}\} \in \mathbb{R}^{B \times D}$ denote the continuous representation of a batch of documents with batch size of $B$. The dot-product results are represented by $\mathbf{H} = \mathbf{D}_t \cdot \mathbf{E}_t^\top \in \mathbb{R}^{B \times K}$. To obtain distinct docids, we calculate an alternative $\mathbf{H}^* = \text{Diag}(\mathbf{u}) \exp(\frac{\mathbf{H}}{\epsilon}) \text{Diag}(\mathbf{v})$, where $\mathbf{u}$ and $\mathbf{v}$ are re-normalization vectors in $\mathbb{R}^K$ and $\mathbb{R}^B$, respectively. The re-normalization vectors are computed via the iterative Sinkhorn-Knopp algorithm [8]. Finally, $\mathbf{H}^*$ is used instead of $\mathbf{H}$ in the $\text{Softmax}$ and $\arg\max$ (Eq. 2) operations to obtain the docid $z_t$.

# 4 Experimental Setup

## 4.1 Datasets and evalutaion metrics

We conduct experiments on three well-known document retrieval datasets: NQ320K [15], MS MARCO [4], and BEIR [38]. We divide the test set of NQ320K into *seen test* and *unseen test*, based on whether the target documents of the query have annotated queries in the training data, to test the generalization ability of the model on unlabeled documents. More details about data pre-processing and data statistics are listed in the Appendix A.

On NQ320K, we use Recall@{1,10,100} and Mean Reciprocal Rank (MRR)@100 as evaluation metrics, following [41]. On MS MARCO, we use Recall@{1, 10, 100} and MRR@10 as evaluation metrics, following [46]. On BEIR, we use nDCG@10 as the main metrics and calculate the average nDCG@10 values across multiple downstream sub-datasets as overall metrics.

## 4.2 Baselines

We consider three types of baselines: sparse retrieval methods, dense retrieval methods, and generative retrieval methods. The **sparse retrieval** baselines are: BM25 [32] and DocT5Query [24]. The **dense retrieval** baselines are: DPR [13], ANCE [42], Sentence-T5 [22], GTR [23], and Contriever [11]. The **generative retrieval** baselines are: GENRE [5], DSI [37], SEAL [1], CGR-Contra [17], DSI-QG [47], NCI [41], and Ultron [46]. The following three baselines use the same pre-trained LM T5 as GENRET: (i) Sentence-T5 outputs continuous vectors, (ii) GENRE outputs document titles, and (iii) DSI-QG outputs clustering IDs, while GENRET outputs docids learned using the proposed tokenization method. See Appendix B for more details on the other baselines.

## 4.3 Implementation details

**Hyper-parameters.** In our experiments, we utilize the T5-Base model [27] as the base Transformer and initialize a new codebook embedding $\mathbf{E}_t$ for each time step. The parameters of both the encoder-decoder and codebook are shared between the tokenization model and the retrieval model. We set the number of clusters to be $K = 512$ for all datasets, with the length of the docid $M$ being dependent on the number of documents present. For datasets containing a larger number of candidate documents, a larger value of $M$ is set to ensure that all documents are assigned unique document ids. In the docid re-assignment, the hyper-parameter $\epsilon$ is set to 1.0, and the Sinkhorn-Knopp algorithm is executed for 100 iterations.

**Indexing with query generation.** Following previous work [47, 41, 40], we use query generation models to generate synthetic (query, document) pairs for data augmentation. Specifically, we use the pre-trained query generation model from DocT5Query [24] to augment the NQ and MS MARCO datasets. In query generation, we use nucleus sampling with parameters $p = 0.8, t = 0.8$ and generate five queries for each document in the collection. For the BEIR datasets, we use the queries generated by GPL [40]. GPL uses a DocT5Query [24] generator trained on MS MARCO to generate about 250K queries for each BEIR dataset. Note that the query generator used for BEIR is purely trained on MS MARCO (without using any training data of BEIR) and thus conforms to the zero-shot setting of BEIR [38, 40].

**Training and inference.** The proposed models and the reproduced baselines are implemented with PyTorch 1.7.1 and HuggingFace transformers 4.22.2. We optimize the model using AdamW and set the learning rate to $5e - 4$. The batch size is 256, and the model is optimized for up to 500k steps for each timestep. During training, we pre-gather documents which share the same docid prefix into a batch. Therefore, the reassignment strategy is applied to a batch, where we aim to have documents with as diverse IDs as possible. We add a factor of 0.1 to the reconstruction losses to balance the scale. In progressive training, we first warm up the model for 5K steps and then initialize the codebook using the clustering centroids as mentioned in Section 3.2.1. We use constrained clustering[4] to obtain diverse clustering results. During inference, we use beam search with constrained decoding [5] and a beam size of 100.

---

[4]`https://github.com/joshlk/k-means-constrained`

Table 1: Results on Natural Questions (NQ320K). The results of the methods marked with $^\dagger$ are from our own re-implementation, others are from their official implementation. * and ** indicate significant improvements over previous-best generative retrieval baselines with p-value $< 0.05$ and p-value $< 0.01$, respectively. ♮ and ♯ indicate significant improvements over previous-best dense retrieval baselines with p-value $< 0.05$ and p-value $< 0.01$, respectively. The best results for each metric are indicated in boldface.

| Method | Full test | | | | Seen test | | | | Unseen test | | | |
|---|---|---|---|---|---|---|---|---|---|---|---|---|
| | R@1 | R@10 | R@100 | MRR | R@1 | R@10 | R@100 | MRR | R@1 | R@10 | R@100 | MRR |
| *Sparse retrieval* | | | | | | | | | | | | |
| BM25 [32] | 29.7 | 60.3 | 82.1 | 40.2 | 29.1 | 59.8 | 82.4 | 39.5 | 32.3 | 61.9 | 81.2 | 42.7 |
| DocT5Query [24] | 38.0 | 69.3 | 86.1 | 48.9 | 35.1 | 68.3 | 86.4 | 46.7 | 48.5 | 72.9 | 85.0 | 57.0 |
| *Dense retrieval* | | | | | | | | | | | | |
| DPR [13] | 50.2 | 77.7 | 90.9 | 59.9 | 50.2 | 78.7 | 91.6 | 60.2 | 50.0 | 74.2 | 88.7 | 58.8 |
| ANCE [42] | 50.2 | 78.5 | 91.4 | 60.2 | 49.7 | 79.2 | 92.3 | 60.1 | 52.0 | 75.9 | 88.0 | 60.5 |
| Sentence-T5$^\dagger$ [22] | 53.6 | 83.0 | 93.8 | 64.1 | 53.4 | 83.9 | 94.7 | 63.8 | 56.5 | 79.5 | 90.7 | 64.9 |
| GTR-Base [23] | 56.0 | 84.4 | 93.7 | 66.2 | 54.4 | 84.7 | 94.2 | 65.3 | 61.9 | 83.2 | 92.1 | 69.6 |
| *Generative retrieval* | | | | | | | | | | | | |
| GENRE$^\dagger$ [5] | 55.2 | 67.3 | 75.4 | 59.9 | 69.5 | 83.7 | 90.4 | 75.0 | 6.0 | 10.4 | 23.4 | 7.8 |
| DSI$^\dagger$ [37] | 55.2 | 67.4 | 78.0 | 59.6 | 69.7 | 83.6 | 90.5 | 74.7 | 1.3 | 7.2 | 31.5 | 3.5 |
| SEAL [1] | 59.9 | 81.2 | 90.9 | 67.7 | - | - | - | - | - | - | - | - |
| CGR-Contra [17] | 63.4 | 81.1 | - | - | - | - | - | - | - | - | - | - |
| DSI-QG$^\dagger$ [47] | 63.1 | 80.7 | 88.0 | 69.5 | 68.0 | 85.0 | 91.4 | 74.3 | 45.9 | 65.8 | 76.3 | 52.8 |
| NCI [41] | 66.4 | 85.7 | 92.4 | 73.6 | 69.8 | 88.5 | 94.6 | 76.8 | 54.5 | 75.9 | 84.8 | 62.4 |
| **Ours** | **68.1**\*♯ | **88.8**\*♮ | **95.2**\* | **75.9**\*♮ | **70.2**♯ | **90.3**♮ | **96.0**♮ | **77.7**♯ | **62.5**\*\* | **83.6**\*\* | **92.5**\*\* | **70.4**\*\* |

## 5 Experimental results

### 5.1 Main results

**Results on NQ320K.** In Table 1, we list the results on NQ320K. GENRET outperforms both the strong pre-trained dense retrieval model, GTR, and the previous best generative retrieval method, NCI, thereby establishing a new state-of-the-art on the NQ320K dataset. Furthermore, our results reveal that existing generative retrieval methods perform well on the seen test but lag behind dense retrieval methods on the unseen test. For example, NCI obtains an MRR@100 of 76.8 on the seen test, which is higher than the MRR@100 of 65.3 obtained by GTR-Base. However, on unseen test data, NCI performs worse than GTR-Base. In contrast, GENRET performs well on both seen and unseen test data. This result highlights the ability of GENRET to combine the advantages of both dense and generative retrieval by learning discrete docids with semantics through end-to-end optimization.

**Results on MS MARCO.** Table 2 presents the results on the MS MARCO dataset. GENRET outperforms generative retrieval methods such as Ultron and dense retrieval baselines such as ANCE and Sentence-T5. Furthermore, previous generative retrieval methods (e.g., GENRE, Ultron) utilizing metadata such as the title and URL, while exhibiting decent performance on the NQ320K dataset, underperform in comparison to dense retrieval and sparse retrieval methods on the MS MARCO dataset. This may be because the NQ320K dataset retrieves Wikipedia documents, where metadata like the title effectively capture the semantics of the document. In the case of the MS MARCO dataset, which is a web search dataset, the metadata often does not adequately characterize the documents, resulting in a decline in performance of the generative retrieval model. In contrast, GENRET learns to generate semantic docids that effectively enhance the generative retrieval model.

**Results on BEIR.** Table 3 lists the results of the baselines and GENRET on six datasets of BEIR. These datasets represent a diverse range of information retrieval scenarios. On average, GENRET outperforms strong baselines including BM25 and ST5 GPL, and achieves competitive results compared to previous-best sparse and dense retrieval methods. Additionally, GENRET demonstrates a significant improvement over the previous generative retrieval model GENRE that utilizes titles as docids. Furthermore, GENRE performs poorly on some datasets, such as BEIR-Covid and BEIR-SciDocs. This may be because the titles of the documents in these datasets do not adequately capture their semantic content.

Table 2: **Results on MS MARCO.** The results of the methods marked with [†] are from our own re-implementation, other results are cited from the original paper or implemented using official code. The best results are indicated in boldface.

| Method | R@1 | R@10 | R@100 | MRR |
|---|---|---|---|---|
| *Sparse retrieval* | | | | |
| BM25 [32] | 39.1 | 69.1 | 86.2 | 48.6 |
| DocT5Query [24] | 46.7 | 76.5 | 90.4 | 56.2 |
| *Dense retrieval* | | | | |
| ANCE [42] | 45.6 | 75.7 | 89.6 | 55.6 |
| Sentence-T5[†] [22] | 41.8 | 75.4 | 91.2 | 52.8 |
| *Generative retrieval* | | | | |
| GENRE[†] [5] | 35.6 | 57.6 | 79.1 | 42.3 |
| Ultron-URL [46] | 29.6 | 67.8 | - | 40.0 |
| Ultron-PQ [46] | 31.6 | 73.1 | - | 45.4 |
| Ultron-Atomic [46] | 32.8 | 74.1 | - | 46.9 |
| **Ours** | **47.9** | **79.8** | **91.6** | **58.1** |

Table 3: **nDCG@10 results on BEIR.** The results of the methods marked with [†] are from our own re-implementation, other results are cited from the original paper or implemented using official code. ST5 GPL denotes Sentence-T5 trained on GPL datasets [40].

| Method | Arg | Covid | NFC | SciF. | SciD. | FiQA | **Avg.** |
|---|---|---|---|---|---|---|---|
| *Sparse retrieval* | | | | | | | |
| BM25 [32] | 29.1 | 58.9 | 33.5 | 67.4 | 14.8 | 23.6 | 37.8 |
| DocT5Query [24] | 34.9 | 71.3 | 32.8 | 67.5 | 16.2 | 29.1 | 41.9 |
| *Dense retrieval* | | | | | | | |
| ST5 GPL[†] [22] | 32.1 | 74.4 | 30.1 | 58.6 | 12.7 | 26.0 | 39.0 |
| Contriever [11] | 40.0 | 68.8 | 33.5 | 61.4 | 16.3 | 30.7 | 41.8 |
| *Generative retrieval* | | | | | | | |
| GENRE[†] [47] | 42.5 | 14.7 | 20.0 | 42.3 | 6.8 | 11.6 | 30.0 |
| **Ours** | 34.3 | 71.8 | 31.6 | 63.9 | 14.9 | 30.2 | 41.1 |

## 5.2 Analytical experiments

We further conduct analytical experiments to study the effectiveness of the proposed method.

In Figure 3 (left), we plot the frequencies of docids at the first timestep of various learning methods. We label each method using a box with a docid and a diversity metric $d$, which is calculated by: $d = 1 - \frac{1}{2n} \sum_{j=1}^{K} |n_j - n_u|$, where $|\cdot|$ represents the absolute value, $n$ denotes the total number of documents, $n_j$ denotes the number of documents that have a docid $= j$, and $n_u = \frac{n}{K}$ is the expected number of documents per docid under the uniform distribution.

The results demonstrate the superiority of GENRET (represented by the yellow line) in terms of distribution uniformity. It uses all the potential docid $k = 512$ and achieves the highest diversity metric with a value of $d = 0.90$. The method without docid reassignment also yields a relatively balanced distribution, with a diversity metric of $d = 0.77$. However, the distribution of the method without diverse codebook initialization is highly uneven, which could be due to the fact that most of the randomly initialized codebook embeddings are not selected by the model during the initial training phase, leading to a lack of update and further selection in subsequent training. Additionally, the models without diverse clustering tend to converge to a trivial solution where all documents are assigned the same docid.

In Figure 3 (right), the results of two ablated variants are presented. First, GENRET w/o learning is a generative model that has been trained directly using the final output docid from GENRET, without utilizing the proposed learning scheme. Its retrieval performance is comparable to that of GENRET on seen test data; however, it is significantly lower on unseen test data. This variant demonstrates that the generative retrieval model jointly trained with auto-encoding objectives can represent documents more sensibly based on semantics. This could enhance performance on the less-optimized documents. Contrarily, the parameters obtained via cross-entropy loss on docid generation tasks are less effective in conveying the semantic information of documents. Secondly, GENRET w/ T5-Small uses a small model, and its performance is inferior to that of GENRET using T5-Base. However, the gap between the performance on seen and unseen test data is smaller, which could be attributed to the limited fitting capacity of the small model.

In Appendix C, we evaluate the proposed model's capacity to retrieve new documents and find that it performs well in adapting to new documents compared to existing document tokenization approaches. Additionally, in Appendix D, we analyze the efficiency of various retrieval models in comparison to the various baselines in terms of memory usage, offline, and online latency, and show the advantages of the proposed model.

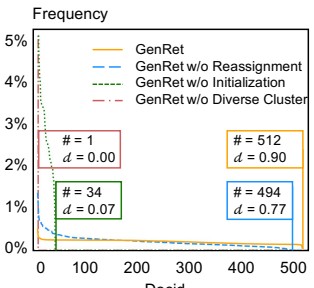 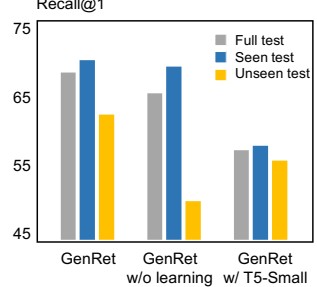

Figure 3: Left: Docid distribution on NQ320K. The id are sorted by the assigned frequency. Right: Ablation study on NQ320K.

## 5.3 Qualitative analysis

Figure 4 (left) illustrates the document content (title) and the corresponding docid generated by GENRET on the NQ320K dataset. We observe that documents with more similar docids tend to have more relevant content. For example, documents with docids starting with 338-173 are related to Email, such as *Email marketing*, *Mail merge*, and *Email address*, while documents with docids starting with 338 relate to information exchange methods (e.g., *Business letter*, *US Postal Service*, and *Postage stamps*), representing a more generalized semantics than Email alone.

Figure 4 (right) illustrates a word cloud of documents grouped by docid prefixes. It is evident that documents within the same group are semantically related. For example, major words for documents with the docid prefix 338 are *mail* and *stamp*. When a second-level docid 173 is added, the corresponding documents become more specifically related to *email*. With the addition of a third-level docid 1, the document group becomes specifically associated with *email marketing* (docid: 338-173-1). Similar patterns can be observed in the other three cases. The case study shows that there is a hierarchical semantic structure within the learned docids.

Figure 5 in Appendix E visualizes the codebook embedding and document embedding. We see that the codebook embedding appears to distribute uniformly within the document representation space, producing meaningful clusters when documents are categorized by docids. We also find that the uniformity of the embedding distribution decreases with increasing docid-length. This may be due to the fact that uniform segmentation is more challenging as the semantic granularity becomes finer.

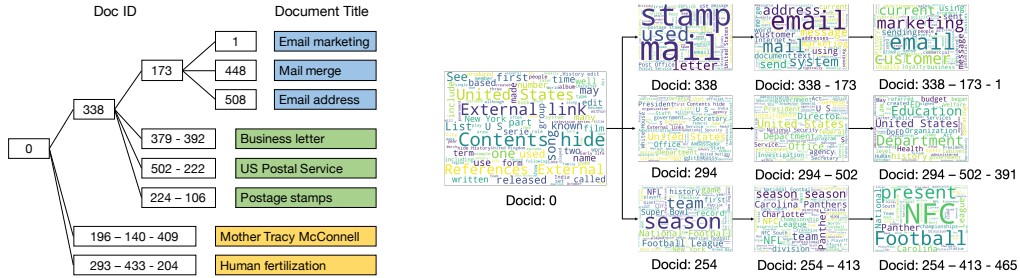

Figure 4: Left: Document titles along with their corresponding docids. It is observed that documents with similar docids tend to have more relevant content. Right: Word cloud representing documents grouped by docid prefixes. This illustrates that different positions of the docid correspond to different levels of information, and the semantics within each cluster are closely related.

## 6 Related work

**Sparse and dense retrieval.** Traditional sparse retrieval calculates the document score using term matching metrics such as TF-IDF [31], query likelihood [16], or BM25 [32]. Sparse retrieval is widely used in practice due to its efficiency, but often suffers from the lexical mismatches [18]. Dense

retrieval (DR) addresses this by presenting queries and documents in dense vectors and calculating their similarities with the inner product or cosine similarity [13]. Various techniques have been proposed to improve DR models, such as hard negative mining [42, 26], late interaction [14, 34], knowledge distillation [10, 19], and pre-training [30, 23, 11]. Despite their success, DR approaches have several limitations [5, 21]: (i) DR models employ an index-retrieval pipeline with a fixed search procedure (MIPS), making it difficult to optimize the model end-to-end [37, 41]. (ii) Training DR models relies on contrastive learning [13] to distinguish positives from negatives, which is inconsistent with large LMs training objectives [3] and fails to fully utilize the capabilities of pre-trained LMs [1, 35].

**Generative retrieval.** Generative retrieval is gaining attention. It retrieves documents by generating their docid using a generative model like T5. Generative retrieval presents an end-to-end solution for document retrieval tasks [37, 21] and allows for better exploitation of the capabilities of large generative LMs [1]. Cao et al. [5] first propose an autoregressive entity retrieval model to retrieve documents by generating titles. Tay et al. [37] propose a differentiable search index (DSI) and represent the document as atomic id, naive string, or semantic string. Bevilacqua et al. [1] suggest using arbitrary spans of a document as docids. Additionally, multiple-stage pre-training [7, 46], query generation [41, 47, 46], contextualized embedding [17], and continual learning [20], have been explored in recent studies. Recently, Tang et al. [36] introduce a query-based docid and the rehearsal-based document indexing to improve DSI. However, existing generative retrieval models have a limitation in that they rely on fixed document tokenization to produce docids, which often fails to capture the semantic information of a document [37]. It is an open question how one should define the docids. To further capture document semantics in the docid, we propose document tokenization learning methods. The semantic docid is generated by the proposed discrete auto-encoding learning scheme in an end-to-end manner. Concurrently, Rajput et al. [28] propose a RQ-VAE module to produce semantic docids for generative recommender systems. As a comparison, our proposed method jointly models the tokenization and retrieval tasks with shared parameters to better align the model's representation of the two tasks and are optimized in an end-to-end manner.

**Discrete representation learning.** Learning discrete representations using neural networks is an important research area in machine learning. For images, Rolfe [33] proposes the discrete variational autoencoder, and VQ-VAE [39] learns quantized representations via vector quantization. Dall-E [29] uses an autoregressive model to generate discrete image representation for text-to-image generation. Recently, representation learning has attracted considerable attention in NLP tasks, for tasks such as machine translation [48], dialogue generation [45], and text classification [12, 43]. For document retrieval, RepCONC [44] uses a discrete representation learning method based on constrained clustering for vector compression. We propose a document tokenization learning method for generative retrieval, which captures the autoregressive nature of docids by progressive training and enhances the diversity of docids by diverse clustering techniques.

# 7 Conclusions

This paper has proposed a document tokenization learning method for generative retrieval, named GENRET. The proposed method learns to tokenize documents into short discrete representations (i.e., docids) via a discrete auto-encoding approach, which ensures the semantics of the generated docids. A progressive training method and two diverse clustering techniques have been proposed to enhance the model's training. Empirical results on various document retrieval datasets have demonstrated the effectiveness of the proposed method. Especially, GENRET achieves outperformance on unseen documents and can be well generalized to multiple retrieval tasks.

The limitations of this work include experiments only on moderately sized datasets like NQ320K. The employed data are in sufficient quantity to validate the effectiveness of the proposed method, but application to larger-scale data may require more model parameters and computational resources. Another aspect for improvement is the generalization of the model to unoptimized document collections in different types or domains, as compared to continuous embedding approaches. We recognize this as an important research question for generative retrieval but believe that addressing this is beyond the scope of this paper. In future work, we would like to extend the approach to large document collections. We also plan to explore generative pre-training for document tokenization using large-scale language models. Additionally, we intend to investigate the dynamic adaptation of docid prefixes for progressive training.

## Acknowledgements

This work was supported by the Natural Science Foundation of China (62272274, 61972234, 62072279, 62102234, 62202271), the Natural Science Foundation of Shandong Province (ZR2021QF129, ZR2022QF004), the Key Scientific and Technological Innovation Program of Shandong Province (2019JZZY010129), the Fundamental Research Funds of Shandong University, the China Scholarship Council under grant nr. 202206220085, the Hybrid Intelligence Center, a 10-year program funded by the Dutch Ministry of Education, Culture and Science through the Netherlands Organisation for Scientific Research, `https://hybrid-intelligence-centre.nl`, and project LESSEN with project number NWA.1389.20.183 of the research program NWA ORC 2020/21, which is (partly) financed by the Dutch Research Council (NWO). All content represents the opinion of the authors, which is not necessarily shared or endorsed by their respective employers and/or sponsors.

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

# A Datasets details

Table 4: Statistics of datasets used in our experiments. The three values split by / on # Test queries denote the number of queries in the full, seen subset, and unseen subset, respectively. In BEIR, all queries in the test set are unseen.

| Dataset | # Docs | # Test queries | # Train pairs |
|---|---|---|---|
| NQ320K | 109,739 | 7,830 / 6,075 / 1,755 | 307,373 |
| MS MARCO | 323,569 | 5,187 / 807 / 4,380 | 366,235 |
| BEIR-Arg | 8,674 | 1,406 | - |
| BEIR-Covid | 171,332 | 50 | - |
| BEIR-NFC | 3,633 | 323 | - |
| BEIR-SciFact | 5,183 | 300 | - |
| BEIR-SciDocs | 25,657 | 1,000 | - |
| BEIR-FiQA | 57,638 | 648 | - |

We conduct experiments on three document[5] retrieval datasets: NQ320K, MS MARCO, and BEIR.

**NQ320K.** NQ320K is a popular dataset for evaluating generative retrieval models [37, 41]. It is based on the Natural Questions (NQ) dataset proposed by Google [15]. NQ320K consists of 320k query-document pairs, where the documents are gathered from Wikipedia pages, and the queries are natural language questions. We follow the evaluation setup in NCI [41] and further split the test set into two subsets: *seen test*, in which the annotated target documents of the queries are included in the training set; and *unseen test*, in which no labeled document is included in the training set.

Note that the NQ320K dataset we utilized has been pre-processed based on the NCI [41] and includes approximately 100k documents. This differs from the DSI approach [37], which processed a version of the NQ320K dataset containing about 200k documents. The distinction lies in the method used to remove duplicate documents. In our case, we eliminated duplicates by comparing document titles, whereas DSI employed URLs. For example, pages `https://en.wikipedia.org//w/index.php?title=Statue_of_Liberty&oldid=804877528` and `https://en.wikipedia.org//w/index.php?title=Statue_of_Liberty&oldid=834310497` are two versions of entity *"Statue of Liberty"*. DSI NQ320K treats them as separate documents, whereas our implementation considers them as a single document. We have found that the content of different versions of the same entity's pages is usually almost identical, with only minor variations, often occurring in later parts of the document. Consequently, we believe that distinguishing between different versions of an entity is beyond the model's capabilities, and that using different versions of the same entity as negative examples in training may hurt model performance.

**MS MARCO.** MS MARCO is a collection of queries and web pages from Bing search. To create the document collections, akin to NQ320k and following [46], we sample a subset of original documents by retaining the top-1 document for each query. We evaluate the models on the queries of the MS MARCO dev set and retrieval on the sampled document subset. We did not split the dev set into *seen* and *unseen* because 84% of the queries in the MS MARCO dev set are unseen.

**BEIR.** BEIR is a collection of datasets for heterogeneous retrieval tasks. In this paper, we evaluate the models on 6 BEIR datasets, which include distinct retrieval tasks and document collections from NQ and MS MARCO: (i) BEIR-Arg retrieves a counterargument to an argument; (ii) BEIR-Covid retrieves scientific articles about the COVID-19 pandemic; (iii) BEIR-NFC retrieves medical documents from PubMed; (iv) BEIR-SciFact retrieves scientific papers for fact-checking; (v) BEIR-SciDocs retrieves citations for scientific papers; (vi) BEIR-FiQA retrieves financial documents. All the queries in BEIR test set are *unseen* [38].

We summarize the statistics of above datasets in Table 4.

---

[5]A document in our context typically corresponds to a webpage, while a passage is a piece of text within a document. Our study focuses on building docid for documents. Using document retrieval tasks allows us to make fair comparisons with existing methods that utilize document URLs, titles, etc. And it also aligns with the settings of NQ320K and BEIR. Passage retrieval, as a different task, could lead to different evaluation results.

# B  Baselines

The sparse retrieval baselines are as follows:

- **BM25**, uses the tf-idf feature to measure term weights; we use the implementation from `http://pyserini.io/`.
- **DocT5Query**, expands a document with possible queries predicted by a finetuned T5 with this document as the input.

The dense retrieval baselines are as follows:

- **DPR** [13], a dual-encoder model using the representation of the `[CLS]` token of BERT.
- **ANCE** [42], an asynchronously updated ANN indexer is utilized to mine hard negatives for training a RoBERTa-based dual-encoder model.
- **Sentence-T5** [22], a dual-encoder model that uses T5 to produce continuous sentence embeddings. We reproduce Sentence-T5 (ST5 for short) on our datasets, the model is based on T5-Base EncDec model and is trained with in-batch negatives.
- **GTR** [23], a state-of-the-art dense retrieval model that pre-trains sentence-T5 on billions of paired data using contrastive learning.
- **Contriever** [11], a dual-encoder model pre-trained using unsupervised contrastive learning with independent cropping and inverse cloze task.

And the generative retrieval baselines are as follows:

- **GENRE** [5], an autoregressive retrieval model that generates the document's title. The original GENRE is trained on the KILT dataset [25] using BART, and we reproduce GENRE on our datasets using T5 for a fair comparison. For datasets without title, we use the first 32 tokens of the document as pseudo-title.
- **DSI** [37], which represents documents using hierarchical K-means clustering results, and indexes documents using the first 32 tokens as pseudo-queries. As the original code is not open source, we reproduce DSI using T5-base and the docids of NCI [41].
- **SEAL** [1] uses arbitrary n-grams in documents as docids, and retrieves documents under the constraint of a pre-built FM-indexer. We refer to the results reported by Wang et al. [41].
- **CGR-Contra** [17], a title generation model with a contextualized vocabulary embedding and a contrastive learning loss.
- **DSI-QG** [47], uses a query generation model to augment the document collection. We reproduce the DSI-QG results using T5 and our dataset.
- **NCI** [41], uses a prefix-aware weight-adaptive decoder and various query generation strategies, including DocAsQuery and DocT5Query. In particular, NCI augments training data by generating 15 queries for each document.
- **Ultron** [46], uses a three-stage training pipeline and represents the document as three types of identifiers, including URL, PQ, and Atomic.

# C  Performance on retrieving new documents

In this experiment, we investigate the impact of various document tokenization techniques on the ability of generative retrieval models to retrieve new documents. The generative models with different tokenization methods are trained on NQ320K data, excluding unseen documents, and are evaluated on NQ320K Unseen test set and BEIR-{Arg, NFC, SciDocs} datasets. For the baseline methods, which use rule-based document tokenization methods, the docids are generated for the target document collection using their respective tokenization techniques. In contrast, our proposed method uses a tokenization model to tokenize the documents in the target collection, producing the docids. However, our method may result in duplicate docids. In such cases, all corresponding documents are retrieved and shuffled in an arbitrary order. The results of this evaluation are summarized in Table 5.

Table 5: Zero-shot evaluation on retrieving new documents with different document tokenization methods. The second column indicates the type of docid, where BERT-HC denotes BERT-Hierarchical-Clustering [37], Prefix-HC denotes Prefix-aware BERT-Hierarchical-Clustering [41], and dAE denotes discrete auto-encoding.

| Method | Docid | **NQ** (R@1) Unseen | **BEIR** (nDCG@10) Arg | NFC | SciDocs |
|--------|-------|------|------|------|---------|
| DSI-Naive[†] [37] | Naive String | 0.0 | 0.1 | 1.0 | 0.1 |
| DSI-Atomic[†] [37] | Atomic | 0.0 | 0.2 | 0.8 | 0.1 |
| GENRE[†] [5] | Title | 6.0 | 0.0 | 2.4 | 0.6 |
| DSI[†] [37] | BERT-HC | 1.3 | 1.8 | 11.1 | 5.9 |
| NCI [41] | Prefix-HC | 15.5 | 0.9 | 4.3 | 1.2 |
| **Ours** | dAE | **34.2** | **12.1** | **12.1** | **12.3** |

Document tokenization methods that do not consider the semantic information of the documents, such as Naive String and Atomic, are ineffective in retrieving new documents without model updating. Methods that consider the semantic information of the documents, such as those based on title or BERT clustering, show some improvement. Our proposed document tokenization method significantly improves over these existing rule-based document tokenization methods. For instance, when the model trained on NQ – a factoid QA data based on Wikipedia documents – is applied to a distinct retrieval task on a different document collection, BEIR-SciDocs, a citation retrieval task on a collection of scientific articles, our proposed document tokenization model still showed promising results with an nDCG@10 of 12.3, which is comparable to those models trained on the target document collection. This suggests that our proposed method effectively encodes the semantic information of documents in the docid and leads to a better fit between the docid and the generative retrieval model.

Table 6: Efficiency analysis.

| Method | Memory | Time (Offline) | Top-$K$ | Time (Online) |
|--------|--------|----------------|---------|---------------|
| ANCE | 1160MB | 145min | 100 | 0.69s |
| GTR-Base | 1430MB | 140min | 100 | 1.97s |
| GENRE | **851MB** | **0min** | 100 | 1.41s |
| | | | 10 | 0.69s |
| DSI | 851MB | 310min | 100 | 0.32s |
| | | | 10 | 0.21s |
| **Ours** | 860MB | 220min | 100 | **0.16s** |
| | | | 10 | **0.10s** |

## D  Efficiency analysis

In Table 6, we compare GENRET with baseline models on MS MARCO (323,569 documents) in terms of memory footprint, offline indexing time (not including the time for neural network training), and online retrieval latency for different Top-K values. We have four observations: (i) The memory footprint of generative retrieval models (GENRE, DSI-QG, and the proposed model) is smaller than of dense and sparse retrieval methods. The memory footprint of generative retrieval models is only dependent on the model parameters, whereas dense and sparse retrieval methods require additional storage space for document embeddings, which increases linearly with the size of the document collection. (ii) DSI and GENRET take a longer time for offline indexing, as DSI involves encoding and clustering documents using BERT, while GENRET requires tokenizing documents using a tokenization model. Dense retrieval's offline time consumption comes from document encoding; GENRE uses titles hence no offline computation. (iii) The online retrieval latency of the generative retrieval model is associated with the beam size (i.e., Top-K) and the length of the docid. GENRET utilizes diverse clustering to generate a shorter docid, resulting in improved online retrieval speed compared to DSI and GENRE.

# E    Embedding visualization

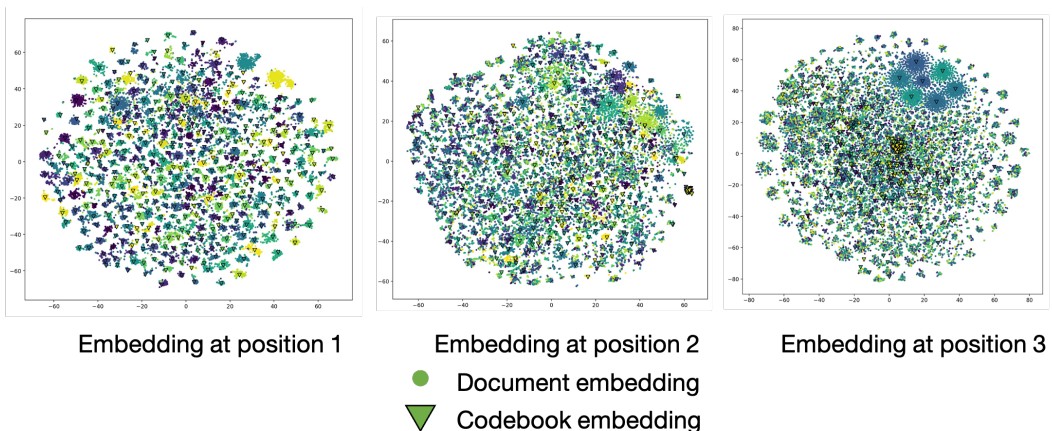

Embedding at position 1          Embedding at position 2          Embedding at position 3

● Document embedding

▽ Codebook embedding

Figure 5: t-SNE visualization of the codebook embedding and document embedding on the NQ320K dataset. The codebook embedding is uniformly scattered in the document representation space.

