# OpenReview forum: "Learning to Tokenize for Generative Retrieval"
_NeurIPS.cc/2023/Conference — NeurIPS 2023 poster_

### Official Review · Reviewer_xNtF · 2023-07-04

**Soundness:** 4 excellent
**Presentation:** 2 fair
**Contribution:** 3 good
**Rating:** 6
**Confidence:** 3

**Summary:**

As one of the mainstream paradigms for document retrieval, generative approaches have enjoyed a steady growth of interest thanks to the recent thriving of large language models. Document tokenization is a crucial step in generative retrieval, which is rule-based in most existing methods, usually generalizing poorly to unlabeled documents. To this end, the paper presents a novel document tokenization learning framework GENRET that learns to tokenize a document into semantic docids in a discrete auto-encoding scheme. GENRET consists of a shared sequence-to-sequence-based document tokenization model, a generative retrieval model, and a reconstruction model, optimized in an end-to-end fashion using a well-designed progressive training scheme to stabilize the training process. Experimental results show that GENRET significantly improves over the baseline methods, especially in generalizing to unseen documents.

**Strengths:**

1. The authors have identified a significant problem in existing generative retrieval approaches. Conventional practice typically treats document tokenization as a fixed pre-processing step; however, this is not optimal as ad-hoc document tokenization methods often fail to capture the complete semantics of a document and generalize poorly to unlabeled documents. Therefore, the paper proposes for the first time to take document tokenization as a learnable module and introduces a novel framework to represent documents as discrete docids that effectively capture the semantic information of the document.
2. The auto-encoding scheme is not new but reasonably integrated into the proposed framework. The progressive training scheme effectively addresses the challenge of learning docids in an autoregressive fashion and stabilizes the training process. The diverse clustering techniques successfully facilitate the diversity of generated docids.
3. Extensive experiments on various benchmark datasets have demonstrated the superiority of the proposed method against established document retrieval approaches.

**Weaknesses:**

1. I think the presentation of the method description (Sec 3) could be further improved, given that some details are not clearly explained. For instance, according to Eq. 4, the reconstruction model is non-parametric, and the stop gradient operator is applied to $\mathbf{d}_t$ and $\mathbf{d}^*_t$, so which part of the parameters the reconstruction loss is aimed at optimizing? I guess it's probably the parameters of the codebook and the encoder-decoder. However, the authors mentioned that "the gradients to document representation $\mathbf{d}_T$ are defined as $\frac{ \partial \mathcal{L}_r}{ \partial \mathbf{d}_T} \coloneqq \frac{ \partial \mathcal{L}_r}{ \partial \mathbf{z}_T}$ " (in Line 173), but $\mathbf{z}_T$ is actually taken from the codebook based on Eq. 5, so the reconstruction loss $\mathcal{L}\_{rec}$ is to optimize the codebook only? Moreover, does the generative retrieval model $P$ share the same codebook with the document tokenization model $Q$? What about the encoder and decoder? In addition, this section also does not cover the inference phase after the training completion. Finally, I do not understand why the docid re-assignment technique can promote the diversity of generated docids.
2. All experimental evidence shows that the proposed approach is effective on purely quantitative measures. However, I do not see any qualitative experiments presented, e.g., by visualization analysis to demonstrate that the codebook embeddings are uniformly scattered in the document representation space or the diversity of generated docids is better than baselines, etc.


**Questions:**

**Q1:** Please take a look at **Weakness1** for suggestions to improve the presentation.
**Q2:** In Eq. 8, is the final loss directly the sum of reconstruction loss, commitment loss, and retrieval loss? Should we set different weighting coefficients for each loss to balance the effects between them?
**Q3:** For the implementation details (Sec 4.3), the authors say that "the length of the docid $M$ is dependent on the number of documents present. For datasets containing a larger number of candidate documents, a larger value of $M$ is set to ensure that all documents are assigned unique document ids". So what $M$ is set to for each of the three datasets used in this paper? Is it possible to suggest how $M$ should be taken for a given number of documents from the perspective of theoretical analysis?

**Limitations:**

As mentioned in the conclusion, the limitations of this work are two-fold. One is the lack of validation of the proposed method on large-scale data, and the other arises from the insufficient generalization of the model to unoptimized document collections in different types or domains.

---

> ### Author Rebuttal · Authors · 2023-08-10
>
> Thanks for your time and valuable comments!
>
> **Presentation of method section**
>
> - Thanks for your suggestions, we will add more explanations to the final paper to improve presentation.
>
> - Firstly, the reconstruction loss optimizes both the encoder-decoder and the codebook. Concretely, $z_T$ is the codebook embedding that is closest in terms of inner-product distance to $d_T$. This computation involves a non-differentiable operation–argmax(·). For optimization of $d_T$ using $L_{rec}$, we employ straight-through gradient estimation [37] that allows for the transmission of the $z_T$ gradient to $d_T$.
>
> - Secondly, the parameters of both the encoder-decoder and codebook are shared between the tokenization model and the retrieval model.
>
> - Thirdly, at the inference stage, the query is entered into the encoder, and it is then processed by decoder+codebook in an autoregressive manner to output docid tokens.
>
> - Lastly, an intuitive explanation of re-assignment technique is that in instances where multiple documents receive the same ID, those with lower relative confidence are re-assigned. This re-assignment directs them towards less popular IDs. This process combined with the commitment loss; results in an increase in the prediction probability of these less frequented IDs, and an overall improvement in the diversity and uniformity of the assigned IDs.
>
> **Qualitative analysis**
>
> - Thanks for your suggestions. We visualize the codebook embedding and document embedding in the newly uploaded PDF files. We see that the codebook embedding does uniformly scatter in the document representation space.
>
> **Loss weighting**
>
> - Thanks for the question. During training, I add a factor of 0.1 to the reconstruction losses to balance the scale. We will add more clarification in our final paper.
>
> **Length of docid**
>
> - Thanks for the question. We list the final choice of hyper parameter M as follow, and we will add more detail our final paper.
> | NQ320K | MSMARCO320K | BEIR-Arg | BEIR-Covid | BEIR-NFC | BEIR-SciFact | BEIR-SciDocs | BEIR-FiQA |
> | --- | --- | --- | --- | --- | --- | --- | --- |
> | 3 | 4 | 2 | 4 | 2 | 2 | 3 | 3 |
>
> - A theoretical analysis suggests that given a certain number of documents $D$, the $M$ value is related to the diversity of ID assignments (e.g., the diversity metric defined on line 285). If the assignment is uniform, then $M \propto log D$. Conversely, if the ID distribution is starkly imbalanced, then may $M \propto D$. In practice, predicting the diversity of assignments is challenging. Observations indicate that the diversity of assignments decreases with an increase in docid length. Consequently, in our study the $M$ value is determined through empirical results, some of which are provided in the table above.
>
> **Limitation**
>
> - As we have discussed in conclusion, scaling the model to larger data typically requires longer docid length, more model capacity (parameters), more computation resources.In addition, we would like to add that: (i) our evaluation scale aligns with previous studies such as DSI and NCI. (ii) our proposed method outperforms existing generative retrieval systems on the unseen subset, and finally, application to larger datasets may pose additional problems e.g. increase in computational resources, which may fall outside the realm this study but could provide an interesting direction for future research.
>
> We hope our response can address your concerns.
>
> [37] Oord, Aäron van den et al. Neural Discrete Representation Learning.

---

> > ### Comment · Reviewer_xNtF · 2023-08-13
> > **No further problems for me**
> >
> > I have read the other reviews and the rebuttal. And I thank the authors for adequately addressing my concerns. For $\mathcal{L}_{rec}$, I now better understand how the encoder-decoder and codebook are optimized. The added qualitative results also seem to support the validity of the proposed method well. I'm willing to raise my **score** to "6" if the authors promise to include these results and analysis in the final submission.

---

> > > ### Author Response · Authors · 2023-08-13
> > >
> > > Thank you for your constructive feedback and for considering a higher score. We promise to include these new results and analysis in our final paper.
> > >
> > > Best regards.

---

### Official Review · Reviewer_BkMu · 2023-07-06

**Soundness:** 3 good
**Presentation:** 4 excellent
**Contribution:** 3 good
**Rating:** 5
**Confidence:** 4

**Summary:**

Current document retrieval systems map documents and queries to doc ids. These docids are assigned randomly, by clustering by using text/attribute information. The authors propose a learned document tokenization scheme where the the semantics of the documents are encoded into learned docids. These docids are used in end-to-end document retrieval methods. The proposed method consists of a retrieval model, a tokenization model and a reconstruction model. All three components are jointly trained. The authors show that on slighgt improvements on many different datasets. They in particular show that thir method generalized much better to unseen data.

**Strengths:**

Recently introduced end-to-end retrieval models are becoming more popular because they result in better performance. Learning better tokenization schemes is indeed a very relevant and important problem. The authors investigate a reasonable approach of using an auto-encoder to obtain doc ids. The 14% increase in performance on unseen data for the NQ dataset does suggest that the proposed method leads to learning better docids.

**Weaknesses:**

The main weaknesses are as follows:
* It is not clear why the autoencoder and the retrieval model are jointly trained. Rajput et. al. [1] also propose using an auto-encoder to learn docids. They use a RQ-VAE to learn docids that are conducive for auto-regressive generation, where each additional token in the docid can be thought of as the residual. In their method the auto-encoder and the retrieval model can be separately learned.
* The evaluation using an unseen dataset is only done with the NQ dataset.
* Most retrieval methods usually use a variant of contrastive loss or a cross-entropy loss. It is not clear why the authors use both and there is ablation for the retrieval loss.

[1] Shashank Rajput, Nikhil Mehta, Anima Singh, Raghunandan H Keshavan, Trung Vu, Lukasz
Heldt, Lichan Hong, Yi Tay, Vinh Q Tran, Jonah Samost, et al. Recommender systems with
generative retrieval. arXiv preprint arXiv:2305.05065, 2023.

**Questions:**

Here are some questions I have:
* How is the 'GENRET w/o learning' model trained? The paper mentions that the proposed learning scheme is not used for  'GENRET w/o learning.' Does this mean the progressive training scheme is not used? I'm trying to understand why GENRET performs so much better on the unseen test set.
* Can you compare GENRET with [1]?
* Have you inspected the learned docids? Do you see similar documents being assigned similar docids? Do you see meaningful clusters arise when you group by docids?
* How important is the commitment loss?
* How much better is GENRET when compared to other baselines on an unseen test set from other datasets? Is the improvement as significant as it is with the NQ dataset? The NQ test set is quite small and dividing it into seen and unseen further reduces the number of samples and the generalization might be more convincing if shown on larger datasets or on more datasets.

[1] Shashank Rajput, Nikhil Mehta, Anima Singh, Raghunandan H Keshavan, Trung Vu, Lukasz
Heldt, Lichan Hong, Yi Tay, Vinh Q Tran, Jonah Samost, et al. Recommender systems with
generative retrieval. arXiv preprint arXiv:2305.05065, 2023.

**Limitations:**

Yes, the authors to discuss limitations in terms of scale of dataset and generalization to different types/domains.

---

> ### Author Rebuttal · Authors · 2023-08-10
>
> Thank you for your insightful review! We will address each of your points in turn.
>
> **Compared with Rajput et al.**
>
> - Thanks for your insightful comment. First, we believe it is beneficial to joint the modeling of tokenization and retrieval tasks on text retrieval tasks. From the perspective of task characterization both tokenization and retrieval tasks aim to convert textual semantics into docid within a shared discrete space.
> Thus sharing parameters can potentially align the model's representation of the two tasks and improve their respective capabilities. From an empirical perspective, the results of the w/o learning variant that only employs docid' results for training a separate retrieval model reveal the merits of joint modeling. On the contrary, consider generative sequential recommenders, which predict the upcoming item IDs based on input interaction history (a sequence of IDs). The differences between the sequential recommendation task and content tokenization could potentially be greater than those in text retrieval, potentially leading to differences in modeling. We expect to do more in-depth analysis in our future work.
>
> - Second, our proposed method also factors in the subsequent token as a residual of the previous one, assimilating this knowledge through the objective in Equation 4.
>
> - Third, we would like to note that the paper from Rajput et al. is still a concurrent work that appeared on ArXiv very close to the NeurIPS submission deadline. As a result, comparative analysis is not pursued yet within our submission paper.
>
> - Finally, we believe our approach is appropriate and effective for text retrieval tasks, and we will incorporate a more comprehensive discussion on Rajput et al.'s insightful study in our final version.
>
> **Unseen test set**
>
> - Thank you for your question. We did not split the test set because 84% of the queries in the MS MARCO test set are unseen, and all the queries in the BEIR test set are unseen [1]. Therefore, the results on MS MARCO and BEIR basically demonstrate the model's ability to retrieve on unseen. We will add clarification to our paper about this regard.
>
> **Why both contrastive loss and cross-entropy loss**
>
> - Since the proposed model jointly learns two objectives: decomposing/quantifying document semantic information into discrete docid, and generating relevant docid for a given query, both the contrastive learning loss for optimizing text representation, and the cross-entropy loss for optimizing generation, are employed. Due to the limited timeframe for a rebuttal, we would like to conduct a more in-depth study on the strengths and weaknesses between the joint-model and the pipeline-model in our future study. We thank you for these insightful comments.
>
> **W/o learning**
>
> - This variant directly employs the final docid assigned by the proposed system and trains a separate DSI model following the practice of DSI-QG. The model is trained without adopting progressive training or contrastive loss. This variant demonstrates that the generative retrieval model jointly trained with auto-encoding objectives can represent documents more sensibly based on semantics. This could potentially enhance performance on the less-optimized documents. Contrarily, the parameters obtained via cross-entropy loss on docid generation tasks might be less effective in conveying the semantic information of documents.
>
> **Qualitative analysis of docid**
>
> - Thanks for your suggestions. In our newly uploaded PDF, we add two qualitative analysis:
>
> - Figure 1 shows a case study of document content and the corresponding docid generated by GenRet. We can see that those documents with more similar docid have more relevant content. For example, we find that documents with docid started with 338-173 are related to Email, while documents with docid started with 338 are related to information exchange methods.
>
> - Figure 2 visualizes the codebook embedding and document embedding. We see that the codebook embedding does uniformly scatter in the document representation space, and meaningful clusters arise when documents are grouped by docids.
>
> - Figure 3 illustrates the word cloud of documents grouped by GenRet produced docid prefix. We can see that different positions of docid represent different levels of information, and the semantics within each cluster are related.
>
>
> **Commitment loss**
>
> - The commitment loss can be divided into two parts: the loss for previous docid token $z_{<t}$, and the loss for docid token at the current training step $z_t$. The former is critical for the model not to forget previous tokens during progressive training; the latter is important for the model to converge on the assignment of docid.
>
> - We have included detailed results of the ablation experiments in a newly uploaded PDF. Without commitment loss for previous tokens, the model's prediction accuracy for docid drops to less than 30% within 5 training epochs. Without commitment loss for the current token, the model's docid variance across epochs stays higher than 5%.
>
> We hope our response can address your concerns.
>
> [1] Thakur, Nandan et al. BEIR: A Heterogenous Benchmark for Zero-shot Evaluation of Information Retrieval Models.

---

> > ### Comment · Reviewer_BkMu · 2023-08-21
> >
> > As the authors say that they are willing to include comparisons to Rajput et. al. [1], I've raised my score.I would also recommend adding the  qualitative analysis of docids.

---

> > > ### Author Response · Authors · 2023-08-22
> > >
> > > Thank you for your insightful feedback and for considering a higher score. We promise to include comparisons to Rajput et. al. [1] in our final paper.
> > >
> > > Best regards.

---

### Official Review · Reviewer_BPS1 · 2023-07-07

**Soundness:** 3 good
**Presentation:** 2 fair
**Contribution:** 3 good
**Rating:** 7
**Confidence:** 3

**Summary:**

GENRET, the proposed model learns to tokenize documents into short discrete representations (i.e., docids) via a discrete auto-encoding approach. Authors develop a progressive training scheme to capture the autoregressive nature of docids and diverse clustering techniques to stabilize the training process.

**Strengths:**

- Very strong results on NQ, MS MACRO
- Beats dense retrieval.
- Nice ablation experiments.

**Weaknesses:**

-	Very handwavy in lines 173-174. Explanation on how argmax is bypassed is unintuitive
-	Doc id reassignment looks to depend upon the documents in the batch. Was any specific batching done?


**Questions:**

- GenRet w/o learning part in lines 297 unclear.
- The paper is very dense in math. Authors may consider writing out some more textual explanations of what is happening.

**Limitations:**

Limitations were properly addressed.

---

> ### Author Rebuttal · Authors · 2023-08-10
>
> Thank you for your constructive comments! We appreciate your feedback and will address each of your points in turn.
>
> **Line 173-174**
>
> - Thanks for your comments and we will add more explanation and clarification in this part in our final paper. Specifically, we employ straight-through gradient estimation that allows for the transmission of the $z_T$ gradient to $d_T$ following *Oord et al. 2017 [37]*, which copies the gradient of $d_T$ directly to $z_T$. We will add more explanations to this section and revise the presentation as you suggested.
>
> **Batching**
>
> - Thanks for your insightful question! We pre-gather documents which share the same docid prefix into a batch. Therefore, the reassignment strategy is applied to a batch, where we aim to have documents with as diverse IDs as possible. We will clarify further on this matter in the final paper for better clarity.
>
> **W/o learning**
>
> - This variant directly employs the final docid assigned by the proposed system and trains a separate DSI model following the practice of DSI-QG. This variant demonstrates that the generative retrieval model jointly trained with auto-encoding objectives can represent documents more sensibly based on semantics. This could potentially enhance performance on the less-optimized documents. Contrarily, the parameters obtained via cross-entropy loss on docid generation tasks are less effective in conveying the semantic information of documents. We will add more explanations of this variant in our final paper.
>
> **Presentation suggestion**
>
> - Thanks for your valuable comment. We will add more explanations in the final paper to increase readability.
>
> [37] Oord, Aäron van den et al. Neural Discrete Representation Learning.

---

> > ### Comment · Reviewer_BPS1 · 2023-08-12
> >
> > - Seen
> > No Score change

---

> > > ### Author Response · Authors · 2023-08-13
> > >
> > > Thanks for your time and valuable comment.
> > >
> > > Best regards.

---

### Official Review · Reviewer_e88A · 2023-07-07

**Soundness:** 3 good
**Presentation:** 3 good
**Contribution:** 3 good
**Rating:** 6
**Confidence:** 4

**Summary:**

This paper introduces GenRet, an auto-regressive retriever that focuses on finding the right clusters (or document ID or document tokenization) approach. Compared to previous generative retriever approaches, GenRet uses three different losses, progressive training, and clustering techniques. Overall, the paper is quite interesting and has the potential to improve the performance of information retrieval systems.

**Strengths:**

Interesting and important problem to work on. The document representation/tokenization for generative retrievers is one of the key problems here. I am glad that there is solid research here.



**Weaknesses:**

Prior generative retrievers use a standard encoder-decoder with a pre-computed document ID. However, GenRet requires several modifications to work properly. While this is not necessarily a problem, I believe that some explanation and ablation studies are needed. For example, the retrieval loss makes GenRet a generalization of dual encoders. If the code-book size is the same as the number of documents in the candidate pool, and the length of the doc ID is 1, then GenRet will reduce to something very close to dual encoders. Therefore, I would like to see more analysis on how to decompose the doc ID and how much it affects generation. This type of study cannot be done in prior approaches because the doc IDs are fixed.


**Questions:**

Could you provide more insights in what way the generative doc id is better than prior pre-computed doc id? Did you find the doc id semantically meaningful?


**Limitations:**

The BEIR results are still behind non-generative retrievers.

---

> ### Author Rebuttal · Authors · 2023-08-10
>
> Thanks for your time and insightful comment. We would like to address your questions in turn.
>
> **Compare to dual-encoder**
>
> - Thanks for your insightful comment. We further compared the models with different values of M and K. The results are in the following table. We find that the performance of models with different configurations are close to the NQ320K. When length of the doc ID is 1, the model will be similar to dual encoder, but with the difference that doc embedding is not obtained by encoding with a doc encoder, but is obtained by training as a codebook parameter.
> | Metric | Model 1 |  Model 2 | Model 3 |
> | --- | --- | --- | --- |
> | K | 109,739 | 2,048 | 512 |
> | M | 1  |  2  |  3  |
> | R@1 |  68.5 | 68.8 |  68.1  |
>
> - Employing a longer docid will reduces model parameters but necessitates more training steps. Utilizing a codebook of length 1 significantly increase the numbers of model parameters, rendering the model unsuitable for large-scale data sets. As advised, we will be incorporating more comprehensive analyses to our final paper.
>
> **Qualitative analysis**
>
> - Thanks for your suggestions. In our newly uploaded PDF, we add two qualitative analysis:
>
> - Figure 1 lists document content and the corresponding docid generated by GenRet on NQ320K dataset, and shows that those documents with more similar docid have more relevant content. For example, we find that documents with docid started with 338-173 are related to Email, while documents with docid started with 338 are related to information exchange methods. Therefore, we find the generated docid semantically meaningful.
>
> - Figure 2 visualizes the codebook embedding and document embedding. We see that the codebook embedding does uniformly scatter in the document representation space.
>
> - Figure 3 illustrates the word cloud of documents grouped by GenRet produced docid prefix. We can see that different positions of docid represent different levels of information, and the semantics within each cluster are related.

---

### Official Review · Reviewer_zz1Z · 2023-07-09

**Soundness:** 2 fair
**Presentation:** 3 good
**Contribution:** 3 good
**Rating:** 5
**Confidence:** 5

**Summary:**

This paper works on an emerging research direction, generative retrieval,  where retrieval is considered as a generating the document ids. This paper proposed to learn a seq2seq model to generate docids from the document. The challenge lies in how to propagate the retrieval loss (from another seq2seq model: query -> relevant docids) to the docid generation model.  The paper proposes several techniques, including commitment loss, iterative optimization, and document reconstruction.

Experiments were done on NQ, MS MARCO and BEIR. Results show superior performance of the proposed method compared to prior generative retrieval approaches as well as standard dense retriever baselines.

**Strengths:**

- It is not straight forward to learn a latent docid from query-document relevance labels. The authors propose several techniques to make enable this.
- Authors adopt diverse baselines
- The paper is clearly written

**Weaknesses:**

My main concern is around the MS MARCO results, which seems very different from other published results. E.g., in the ANCE paper, ANCE's MRR is 0.33 [1]; docT5query's MRR is 27.2 [2]. The MRR numbers reported in this paper is ~50-60, and docT5query is better than ANCE according to this paper which is not true based on prior research. This makes me question the soundness of this set of experiments.

For BEIR experiments, it is unclear how the six datasets are selected from all BEIR tasks.


[1] Approximate Nearest Neighbor Negative Contrastive Learning for Dense Text Retrieval. Xiong et al.
[2] https://cs.uwaterloo.ca/~jimmylin/publications/Nogueira_Lin_2019_docTTTTTquery-v2.pdf

**Questions:**

- Please explain why the MS MARCO results are off from results reported by prior work, and why dense retrieval (ANCE) is worse than docT5query on this set up.
- How was the six BEIR dataset picked?

**Limitations:**

Would be nice to discuss how this method scales up to larger corpus, and whether / how it supports index updating.

---

> ### Author Rebuttal · Authors · 2023-08-10
>
> Thanks for your time and insightful comment. We would like to address your questions in turn.
>
> **Two reasons for different numbers for ANCE/docT5query**
>
> - First, please note that we are focusing on the document retrieval task in our paper, whereas the results you referenced are from the passage retrieval task.
>
> - Second, our setting is consistent with existing generative retrieval studies [1,2], where we retain the top-1 document for each query within our document collection. In contrast, the papers you cited utilize the MS MARCO leaderboard passage collection, which maintains the top-10 Bing Search passages for each query [3].
>
>
> **Higher results of docT5query**
>
> - We accredit the differences in results to the different retrieval tasks, and the size and construction methods of doc/passage collection. To provide more implementation details, for ANCE we use the [ance-firstp](https://huggingface.co/sentence-transformers/msmarco-roberta-base-ance-firstp). For docT5query we use [doct5query](https://huggingface.co/castorini/doc2query-t5-base-msmarco).
>
>
> **BEIR data selection**
>
> - Our selection of the six datasets from BEIR is based on the corpora size. In particular, the document size of selected data is around or less than 320K documents, intending to test the method's effectiveness on moderate-sized corpora following previous studies [1,2].
>
> **Scaling corpus and updating index**
>
> - Thanks for the questions! As we have discussed in conclusion, scaling the proposed model to larger corpora typically requires longer docid length, more model capacity (parameters), and more computation resources. In addition, expanding to a larger corpus may confronts several known challenges [4], such as knowledge forgetting and training efficiency, which need to be addressed in our future research.
>
> - We anticipate that the proposed semantic docid generation method will indeed be conducive to large corpora. This is because a more meaningful docid results in a more efficient compression of the document's semantics, thereby reducing the burden on the model to memorize the sequence of these IDs. In comparison, when the docid is arbitrary, eg native string, the model needs to memorize these ID sequences without the aid of meaningful association, leading to additional overhead. We observe relevant evidence in the rate of training loss decline, e.g., the loss declines faster when using a meaningful docid.
>
> - As for updating the index, we presume that expanding docid could provide a feasible solution e.g., adding new docid bits for new documents to differentiate them from existing ones, and continue training the model based on progressive training. In addition, the MOE approach might also be feasible, e.g., indexing documents in separate experts and merging the results of multiple experts through aggregation techniques. We would like to discuss this further and do more exploration in our final version.
>
> We hope our response can address your concerns.
>
> [1] Transformer Memory as a Differentiable Search Index.
> [2] Ultron: An Ultimate Retriever on Corpus with a Model-based Indexer
> [3] MS MARCO: A Human Generated MAchine Reading COmprehension Dataset.
> [4] How Does Generative Retrieval Scale to Millions of Passages?

---

> > ### Comment · Reviewer_zz1Z · 2023-08-22
> >
> > Thank you for the detailed response! They addressed many of my concerns and I'm willing to raise my rating. Please explain in the paper why document retrieval is focused instead of passage retrieval.

---

> > > ### Author Response · Authors · 2023-08-22
> > >
> > > Thank you for your insightful feedback and for considering a higher score.
> > >
> > > A document in our context typically corresponds to a webpage, while a passage is a piece of text within a document. Our study focuses on building docid for documents. Using document retrieval tasks allows us to make fair comparisons with existing methods that utilize document URLs, titles, etc. And it also aligns with the settings of NQ320K and BEIR. Passage retrieval, as a different task, could lead to different evaluation results.
> > >
> > > Thanks for your comments and we will incorporate an explanation in our final paper. We are also open to exploring other IR tasks, e.g., passage retrieval, in our future work.
> > >
> > > Best Regards.

---

### Author Rebuttal · Authors · 2023-08-10

To all.

We appreciate all the reviewers for the constructive comments. We have included four figures in our newly updated PDF:

- Figure 1 presents a case study of document content matched with the relevant docid generated by GenRet. It suggests that documents with similar docids share closely related content. For instance, documents beginning with the docid 338-173 pertain to Email, while those starting with 338 concern various information exchange methods.

- Figure 2 shows a t-sne visualization of both codebook embedding and document embedding on NQ320K. The codebook embedding appears to distribute uniformly within the document representation space, producing meaningful clusters when documents are categorized by docids.

- Figure 3 shows a word cloud composed of documents grouped by GenRet-produced docid prefixes. It’s evident that documents of same group are semantically related.

- Figure 4 illuminates the results of an ablation experiment on commitment loss. The data suggests that commitment loss is beneficial for preventing the model from forgetting previous tokens and aids in enhancement of convergence.

We will add these results to our final paper.

Best regard.

---

### Decision · Program_Chairs · 2023-09-21

**Decision:**

Accept (poster)

**Comment:**

This is a well-written paper that proposes a straightforward but highly effective way of learning document identifiers for generative information retrieval (one of the most important problems in this area). The experimental results are strong, comprehensive, and convincing.

The reviewers raised a few concerns about this paper, but the most critical aspects of those were adequately addressed during the rebuttal.

Given that the strengths of this paper clearly outweigh its weaknesses, this paper is suitable for publication.

The authors are strongly encouraged to carefully consider all of the reviewer feedback, including the rebuttal-related discussion, and to take meaningful steps to incorporate it into the final version of their paper.